# The Clinical Application of Porous Tantalum and Its New Development for Bone Tissue Engineering

**DOI:** 10.3390/ma14102647

**Published:** 2021-05-18

**Authors:** Gan Huang, Shu-Ting Pan, Jia-Xuan Qiu

**Affiliations:** Department of Oral and Maxillofacial Surgery, The First Affiliated Hospital of Nanchang University, Nanchang 330006, China; hg3946@163.com (G.H.); panshuting314@126.com (S.-T.P.)

**Keywords:** porous tantalum, clinical application, additive manufacturing, surface modification, bone tissue engineering

## Abstract

Porous tantalum (Ta) is a promising biomaterial and has been applied in orthopedics and dentistry for nearly two decades. The high porosity and interconnected pore structure of porous Ta promise fine bone ingrowth and new bone formation within the inner space, which further guarantee rapid osteointegration and bone–implant stability in the long term. Porous Ta has high wettability and surface energy that can facilitate adherence, proliferation and mineralization of osteoblasts. Meanwhile, the low elastic modulus and high friction coefficient of porous Ta allow it to effectively avoid the stress shield effect, minimize marginal bone loss and ensure primary stability. Accordingly, the satisfactory clinical application of porous Ta-based implants or prostheses is mainly derived from its excellent biological and mechanical properties. With the advent of additive manufacturing, personalized porous Ta-based implants or prostheses have shown their clinical value in the treatment of individual patients who need specially designed implants or prosthesis. In addition, many modification methods have been introduced to enhance the bioactivity and antibacterial property of porous Ta with promising in vitro and in vivo research results. In any case, choosing suitable patients is of great importance to guarantee surgical success after porous Ta insertion.

## 1. Introduction

Named after the Greek mythological figure Tantalus [1], tantalum or Ta is a rare, rigid and ductile metal element with an extremely high melting point (3017 °C) [2] and density (16.6 g/cm^3^) [3,4]. Ta has excellent biocompatibility and corrosion resistance, has been used in pacemaker electrodes, suture wire, cranioplasty plates, radiopaque markers, foil and mesh for nerve repair since the 1940s [5]. In addition, Ta has been used as single or composite coating material to modify the biological and mechanical properties of pure titanium(Ti) [6,7,8], Ti alloy (Ti6Al4V) [9], polyetheretherketone (PEEK) [10], cobalt-chromium (CoCr) alloy [11,12], magnesium-based alloy [13], pure Fe [14] and 316 L stainless steel [15]. Recently, the advent of Ti-Ta alloy with different Ta element contents indicates a novel means to fabricate implants for bone defect restoration with improved mechanical strength, and satisfactory elastic modulus and biological properties, compared to pure Ti and Ti alloy [2,16,17].

Though lacking intrinsic antibacterial properties [16], Ta has a lower bacterial adhesion level and colonization compared to titanium (Ti) and stainless steel due to the spontaneously formed oxide surface layer (Ta_2_O_5_) [17,18]. The Ta_2_O_5_ layer also has been proven to facilitate the deposition of bone-like apatite coating in simulated body fluid (SBF) [19], and further accelerate the adherence of osseous and soft tissues [20]. Moreover, nanoparticles released from Ta implants have been certified to stimulate the proliferation of osteoblasts via autophagy, and the osteogenic process can further be enhanced by autophagy inducer [21]. Although the osteogenic signaling pathways of Ta have yet been fully explicated, several studies have focused on the TGF-β/Smad3 [22], BMP2/Smad1 [23], Wnt/β-catenin [22,24], Integrin α5β1/ERK1/2 [25,26] and MAPK/ERK pathways [27] that may be involved in the osteogenic effects of Ta. It is also reported that Ta can enhance the osteogenesis of diabetic rabbits by suppressing the activation of ROS-mediated P38 MAPK signaling pathway [28]. Furthermore, Ta upregulates the expression level of osteoprotegrin (OPG) and reduces that of RANKL, which means Ta also can inhibit the activity of osteoclasts [23]. The relative molecular mechanism for the osteogenic effects of Ta has been illustrated in Figure 1.

Compared to its solid counterpart, currently commercialized porous Ta possesses modified physical properties including high porosity (range from 75% to 85%), dodecahedral cell structure and pore sizes ranging from 400 to 600 μm. It has been reported that scaffolds with an average pore size of up to 400 μm and porosity of up to 70% can facilitate cell migration, proliferation, osteogenic differentiation, and blood vessel and bone tissue formation [29,30,31,32]. In this regard, the higher pore size and porosity of porous Ta can also contribute to bone and soft tissue ingrowth due to its extensively three-dimensional inner space and high pore interconnectivity [33,34]. Meanwhile, the high porosity of porous Ta ensures desirable permeability for vascularization and nutrient flow, which can guarantee rapid osteointegration at an early stage [35]. Combined with the inherent high wettability and surface energy, porous Ta can further facilitate the adhesion, differentiation and spread of stem cells [36], osteoblasts [37,38] and chondrocytes [39], as well as vascularized fibrous tissues [40] and tendon [41]. Furthermore, bone ingrowth can be found within the pores of porous Ta as early as 4 weeks after implantation [38] (Figure 2). Many in vivo studies also have highlighted its early osteointegration and evidenced bone ingrowth within the inner pores with Haversian remodeling in the long term [22,41,42,43]. In vitro, after being cultured on the porous Ta, osteoblasts obtained from old female patients (>60 years) showed better proliferation and osteogenesis than those cultured on Ti fiber mesh [44], indicating the potential efficacy of porous Ta for the treatment of patients suffering from osteoporosis.

As shown in Table 1, the mechanical properties of porous Ta can be modified to be more suitable for bone-tissue regeneration, especially for load-bearing parts of the body, via various technology due to its elastic modulus and compressive strength being much more comparable to either cortical or cancellous bones [20,33]. The satisfactory elastic modulus of porous Ta is of great importance to proportionally distribute load stress to adjacent osseous tissues, minimize stress shield effect, prevent bone resorption, and further preserve the adjacent bone stock [46]. In addition, the high friction coefficient of porous Ta also promises primary stability for the porous Ta-based implants or prostheses [42]. It is worth noting that higher pore size and porosity are associated with fine biological performance, but the mechanical strength is the opposite [43]. Therefore, attaining a balance between biological and mechanical properties of porous Ta by adjusting a rational pore size/porosity ratio is a critical challenge for future manufacturing and application.

The commercially available porous Ta implants fabricated via Chemical Vapor Deposition (CVD) by Zimmer Biomet Inc. (Warsaw, IN, USA), also known as trabecular metal, resemble cancellous bone due to their microstructure [33]. Meanwhile, many manufacturers, e.g., Runze Pharmaceutical Co., Ltd. (Chongqing, China) [49] and Printing Additive Manufacturing Co., Ltd. (Zhuzhou, China) [54], have also engaged in the manufacture of porous Ta. At present, additive manufacturing (AM), also known as 3-Dimensional printing or rapid prototyping, has been exploited to fabricate porous tantalum scaffolds or implants. The procedures of AM technology mainly include electron beam melting (EBM), laser engineering net shaping (LENS), and selective laser melting (SLM). Compared with CVD or other traditional subtractive manufacturing, AM exhibits superior performance with satisfactory cost-efficiency, less time and material consumption [53]. With the help of AM technology, both the macrostructure and microstructure of porous Ta can be precisely controlled, during the producing process, according to the design parameters. The Additive manufactured porous Ta scaffolds also show satisfactory fatigue strength and load-bearing capacity [56]. Moreover, many modification methods have been employed to enhance the bioactivity and antibacterial property of porous Ta for its future application in bone tissue engineering.

So far, porous Ta-based implants or prostheses have been extensively applied in orthopedics and dentistry (Figure 3, and typical products are shown in Figure 4). Therefore, the aim of this research is to review the clinical application of porous Ta-based implants or prostheses which have been implemented in orthopedics and dentistry, and summarize new manufacturing and modification methods for this promising porous biomaterial.

## 2. Clinical Application of Porous Ta in Orthopedics and Dentistry

### 2.1. Femoral Head Osteonecrosis

Osteonecrosis of the femoral head can be an extremely harmful disease for young and middle-aged patients who are physically active [61,62]. Therefore, appropriate measures should be taken at an early stage to preserve the femoral head before the final collapse of the femoral head and subchondral plate.

Core decompression has been applied in the salvage of the femoral head for many years, but the lack of mechanical support to the subchondral bone after debridement of the necrotic bone may further result in the collapse of the head [63]. Meanwhile, in their histopathology study, Gonzalez Del Pino et al. [64] found that the new bone formation originated mainly from the host bones rather than the vascularized grafts. In this regard, as a reasonable substitute for vascularized fibular autografts, porous Ta rod has been used as a supplementary approach to sustain the bony defect portion after core decompression [65].

Primarily designed to sustain the structure of the subchondral plate and stimulate osteogenesis of the host bone, porous Ta rod has been proven to alleviate the deterioration of femoral head necrosis and postpone the final conversion to total hip arthroplasty, in the majority of publications, for early or intermediate stage patients [65,66,67,68]. Although the efficacy of this tantalum rod remains controversial in the long-term [69], removal of the rod would be an obstacle during conversion to total hip arthroplasty [70]. The survival rates after porous Ta rod insertion is impacted by multiple factors including the stage of the disease, corticosteroid usage, osteonecrosis lesion volume and location, bone marrow edema, and joint effusion [71,72]. The presence of bone marrow edema has been proven to be a poor prognostic factor of femoral head osteonecrosis and also a predictor of conversion to total hip arthroplasty (THA). Furthermore, patients with bone marrow edema had a significantly higher likelihood of eventually resorting to THA [72].

It should be noted that the diameter of a porous Ta rod is only 10 mm, which confines the supporting area of the rod; if the lesion size was larger than that diameter, collapse would occur at other areas [73]. Moreover, the histopathological analysis of 15 retrieved porous Ta rods revealed 1.9% bone ingrowth, and mechanical support for the subchondral bone was proven to be insufficient [74]. Thus, improvements in implant design and surgical technique are needed, and the patients’ necrotic stages should also be scrutinized before the surgical procedure is undertaken [75]. Accordingly, many modified surgical techniques have been introduced to enhance the osteogenesis ability of porous Ta rod, including a combined technique involving bone marrow aspired from iliac crest [58], combination with vascularized bone grafting alone [76], or with bone marrow mesenchymal stem cells (BMMSCs) and vascularized autografts [77]. However, longer-term follow-up clinical trials are still desired to verify the efficacy of these modified methods.

### 2.2. Hip Arthroplasty

The porous Ta acetabular cup for primary THA is fabricated by directly compressing the ultra-high molecular polyethylene into an elliptical porous Ta shell. This kind of monoblock acetabular component design has theoretically diminished the occurrence of backside wear, and the absence of screw holes prevents the access of polyethylene wear debris, which can infiltrate the bone–implant interface, and which has long been regarded as an initiating factor resulting in aseptic loosening of the cup [78]. The porous Ta shell with low elastic modulus, high friction coefficient and excellent osteoconductivity can help to preserve or even increase the bone stock of adjacent acetabulum and, if necessary, facilitate the revision surgery [79].

In a preclinical research, 22 porous Ta acetabular components were exploited in a canine model [37]. The results revealed that the bone ingrowth depth of the 22 cups ranged from 0.2 to 2 mm after 6 months. Furthermore, the average bone ingrowth was 16.8% in all sections and 25.1% in the periphery; both were better than the results of another canine model study in which bone ingrowth in titanium fiber and Co-Cr was 21.5% and 13.4%, respectively [80]. Clinically, 151 hips were followed up for 8–10 years post primary THA in a prospective study [81]. Although periacetabular gaps with lengths ranging from 1 to 5 mm could be found in 25 hips at early stage, those gaps disappeared after 24 weeks. The follow-up radiograph verified the absence of radiolucency, osteolysis of the adjacent bone, polyethylene wear debris and cup loosening. All these indicated the design advantages of the porous Ta cup. Substantial bone deposition could be found on the surface of a retrieved acetabular component after 50 months due to dislocation in this study. However, the lack of screw holes of the cup may have hampered the direct observation of dome contact during surgery and the final seating of the cup into acetabular socket could not be accurately ensured [81].

As for revision THA, it is a surgical challenge to reconstruct acetabulum with huge bone defects and to restore the primary stability, rotational center and maximal bone–implant contact [82]. Porous Ta acetabular prostheses has been revealed as an optimal option to cope with these formidable challenges [83,84,85,86,87]. The modular design of the porous Ta revision prosthesis provides augmented or buttressed sections to be screwed onto the supra-acetabulum for bone defect reconstruction; subsequently, the elliptical cup is implanted in the acetabular socket against the section with cement laying at the interface of the two components [88]. Many short and medium-term studies have shown promising results of the modular porous Ta acetabular shell and augmentation in the treatment of acetabular dome defects with or without osteolysis of ischium, teardrop and Kohler line disruption (Paprosky type II or III) [85,87,89,90,91,92,93,94]. A ten-year follow-up after revision surgery with porous Ta shell and augmented implantation was conducted by Löchel et al. [95]. The survival rate of 51 patients (53 hips) who had completed the follow-up was 92.5%, with a significant increase in Harris Hip Score (55 before surgery vs. 81 post surgery) after the revision surgery. Meanwhile, the authors strongly recommended the application of screws toward the load transferring and inferior direction in every patient with acetabular defects to stabilize the shell and augmentation, diminish the fretting at the interface of shell–host bone or shell–augmentation and guarantee the primary stability of the long-term survival rate [95]. In addition, porous Ta acetabular implants and augmentations were suggested in the reconstruction of the hip joint after resection of peri-acetabular tumors, in order to ensure satisfactory clinical results at early stage [96]. The irradiated pelvis was reported to always be associated with high aseptic loosening rates of acetabular components [97,98,99]. Even so, porous Ta cups still obtained satisfactory results in THA treatment of irradiated pelvis owing to their high friction coefficient and porous microstructure, as well as rapid bone ingrowth rate [97,98,99]. However, it is imperative to note that transverse acetabulum fracture may occur during or after the revision surgery if excessive reaming is performed to insert large cups (average 58 mm) during the operation [100,101].

### 2.3. Knee Arthroplasty

Porous Ta prosthesis for keen primary and revision reconstruction comprises the monoblock tibial component, the tibial or femoral cone and augmentation, as well as the patella prosthesis. The design of the monoblock tibial component for primary arthroplasty is similar to that of the monoblock acetabular component, with the polyethylene directly compressed into a porous Ta baseplate, which also eliminates the potential occurrence of wear debris infiltrating into bone–implant interface. The mechanical and biological properties of porous Ta guarantee the primary stability of the tibial component and ensure its long-term survival rate [102]. Several short and long term results have shown encouraging efficacy of this cemented or uncemented monoblock tibial component for the treatment of relatively young and active patients [103,104,105,106,107,108]. A histological analysis of a retrieved porous Ta tibial component from a chronically infected knee prosthesis revealed significant bone ingrowth in the posts and post–baseplate interface rather than baseplate, suggesting that fine bone–implant integration could still be obtained even in the infected environment [109]. However, caution should be taken with patients who have heavy weight (average 241.9 lbs) and tall height (average 71.8 inch) and have previously received total knee arthroplasty (TKA) with cementless porous Ta tibial prostheses, as this patient group may easily encounter early medial collapse due to the overload cyclically posed on the medial portion of the tibial prosthesis [110].

Severe distal femoral and proximal tibial bone defects are the greatest challenge in revision total knee arthroplasty. Without adequate bony support and inferior bony structure, the collapse of the tibial or femoral component will inevitably occur. Therefore, porous Ta cones for substitution of tibial and femoral metaphyseal bone defects have been introduced to function as structural grafts, to enhance bone stock, and to regain normal articular alignment with multiple flexibilities for different sizes and positions of bone loss [102,111]. The results of a 5-year study reported by Potter et al. [112] indicated that porous Ta femoral cones could effectively fill the metaphyseal defects of the distal femur and sustain the femoral component after revision TKA. Another five to nine year follow-up study supported the efficient application of porous Ta tibial cones for the restoration of huge osseous loss and facilitated early weight-bearing [113]. However, long-term and comparative analysis is still needed to further verify the viability of these porous cones for massive metaphysis defect reconstruction, and the high price per cone (approximately $4.000) would impede their clinical application at a large scale [114].

Restoring the normal function and structure of the patellofemoral joint will be an integral portion in TKA or revision TKA if the extensor mechanism has been impaired due to patellar resection or severe osseous deficiency. Owing to its capability to favor soft tissue and bone ingrowth [38,40], porous Ta patellar prosthesis has been used to reconstruct the fulcrum role of patella [115]. However, the stability of this novel patellar prosthesis depends mainly on the residual bone stock of patella, rather than soft tissue [116]. Moreover, abundant bone–implant contact and blood supply to the residual patella are critical factors for the long term success of porous Ta patellar prosthesis [117]. Therefore, prudent selection of proper patients should be the prior step before definite surgery is performed, so as to avoid the recurrence of complications such as persistent pain, weakened extensor mechanism, and patellar shell fracture.

### 2.4. Shoulder Arthroplasty

Glenoid component loosening is always a disturbing complication of total shoulder arthroplasty (TSA) or reverse total shoulder arthroplasty (RTSA) despite various methods having been implemented to address it [118,119,120,121,122]. Based on the successful experience in hip and knee arthroplasty, the porous Ta baked monoblock glenoid component has been introduced for TSA [123] and RTSA [124], utilizing the properties of rapid osteointegration and high friction coefficient of porous Ta, which can elongate survivorship of the shoulder prosthesis in the long-run.

Budge et al. [123] reported that prosthetic fracture of the first generation of the porous Ta glenoid component occurred in 4 of 19 patients after 30 months. All four glenoid component fractures appeared at the interface of the keel and the metal plate, indicating a combined effort of cyclic loading and insufficient bony support to the glenoid portion of the prosthesis that finally ignited the fatigue failure at the junction of the keel and the metal plate [123]. Therefore, emphases should be placed on gaining a compacted press-fit pattern of the metal component in surgery [125]. Reinforced with anterior-posterior keels and extended interdigitating at the interface of polyethylene tray and porous Ta plate, the redesigned second generation of porous Ta glenoid prosthesis is introduced to cope with the annoying keel–glenoid plate junction fracture that frequently occurred in the first generation [126].

Improved clinical performance of the second generation porous Ta glenoid component was revealed in 40 patients who were followed up for 38 months post TSA [126]. Significant progress of shoulder function scores were found in all 40 patients without conspicuous prostheses loosening, migration or fracture [126]. After 76 shoulders were replaced with porous Ta glenoid components and followed up for an average of 43.2 months, Panti et al. [127] also reported satisfactory clinical results in terms of improved range of motion, pain relief and advanced function scores with the absence of severe complications such as implant fracture or loosening. The mechanism behind the successful results obtained in the two aforementioned studies mainly depended on the amplified cruciform pegs, which guaranteed the compact press-fit mechanism of the glenoid component in the host bone without cement, and effectively withstood the eccentric loading force that was exerted. Meanwhile, the expanded interlocking between polyethylene tray and porous metal plate helped to resist debonding force and diminish the possibility of potential glenoid component fracture.

Nevertheless, it should be noted that the monoblock porous Ta glenoid component may cause trouble when revision TSA is to be performed, for it would be difficult to be removed due to the stable bone–implant interlocking and substantial bone ingrowth even associated with infected host bone [128], and may subsequently result in central osseous defect or scapular fracture [125]. Still, accumulated metallic debris deposition derived from the porous Ta component with a time-dependent pattern can still be found after anatomic shoulder arthroplasty, which means that mechanical failure remains a threat in terms of long-term survival rate, and that regular radiographic follow-up should be implemented to verify the stability of the porous Ta glenoid component [129].

Superior migration of greater tuberosity is a critical factor, which often results in surgical failure after shoulder hemiarthroplasty in the treatment of complex proximal humeral fractures. The migrated great tuberosity is the main cause of subacromial impingement, shoulder joint stiffness and persistent pain [130]. Humeral stem prostheses fabricated with porous Ta have been proposed to accelerate the anatomic union of greater tuberosity, which can effectively minimize the occurrence of greater tuberosity malposition after surgery and ensure eventual surgical efficacy [131,132].

In addition, porous Ta glenoid augmentations have recently been used to correct glenoid retroversion, with satisfactory efficacy, in 10 patients undergoing repositioning of the retroverted glenoid to a neutral position [133]. Glenoid retroversion caused by dysplasia or degenerative deformity can lead to eccentric loading, permanent posterior subluxation of the humeral head and severe prosthetic failure [133]. Therefore, porous Ta glenoid augmentation can be an optimal approach to correct glenoid deformity, though more evidence is desired to verify its exact efficacy.

### 2.5. Spine Intervertebral Fusion

The efficacy of porous Ta cages applied in anterior cervical spine fusion has been confirmed in a prospective randomized controlled clinic trial conducted by Fernández-Fairen et al. [134]. Compared with traditional autologous iliac bone graft combined with anterior plating, the porous Ta cage insertion group showed an equivalent fusion rate (89% vs. 85%) and post-surgery stability at the end point of a 2-year follow-up, but additional fixation and graft harvesting-related injuries no longer occurred [134]. After 11 years of follow-up, the clinical and radiological results of patients who had received single porous Ta cage insertion for interbody fusion remained satisfactory despite the subsidence of implants, presented within 2–3 mm, occurring without significant complication in 12 patients [135]. Furthermore, several observational studies had also affirmed the efficacy of porous Ta in terms of interbody fusion rate, low complication rate and improved short or long term post-surgery evaluation scores including the SF-36, neck disability index (NDI) and visual analog score (VAS) [136,137,138].

On the contrary, another prospective randomized multicenter study showed a frustrating fusion rate of the stand-alone porous Ta device insertion group compared to the iliac crest autograft group (44% vs. 100%) after a 2-year follow-up [139]. In addition, the histological analysis of two retrieved tantalum blocks from patients diagnosed as nonunion at 6 months and 12 months, respectively, revealed substantial fibrous ingrowth instead of bony tissues [139]. Similarly, Löfgren et al. [140] found a significantly lower fusion rate of porous Ta than iliac crest bone grafting (69% vs. 92%). Considering potential nonunion, Wigfield et al. [141] terminated their prospective study prematurely due to radiolucent lines appearing at the anterior–inferior border of porous Ta implants in four patients, implanted with either blocks or rings, at 6 weeks, though the lucent lines disappeared 12 months post-surgery and the final outcomes of the porous Ta insertion group were better than the autologous bone grafted group.

Accordingly, two meta-analyses [142,143] have recently analyzed the intro-operation and post-surgery parameters including operating time, blood loss, hospital stay, fusion rate, NDI and VAS scores, as well as satisfaction and complication rates of relative clinical trials. Through the two meta-analyses, it has been established that porous Ta implants possess the same efficacy and safety in the surgical treatment of anterior cervical degenerative disc diseases as autologous iliac bone grafting, which has long been regarded as the gold standard [142,143]. However, more randomized controlled trials with large samples are still desired to reinforce the clinical evidence of porous Ta implants.

In addition to a high friction coefficient, porous Ta can offer suitable conditions for rapid bone ingrowth [34], which further guarantees its long-term stability for lumbar intervertebral fusion either with or without the augmentation of pedicle screws [144]. Lequin et al. [145] used standalone porous Ta cages in the treatment of 26 patients suffering from recurrent lumbar disc herniation. Though moderate satisfactory clinical results were revealed and significant relief of back and leg pain was only reported in 46% of patients, 85% of patients showed remarkable improvement in their working-status at 1 year post-surgery [145]. Meanwhile, Lebhar et al. [146] and Butler et al. [57], respectively, reported reliable clinical, functional and radiographic results after the application of porous Ta implant for lumbar intervertebral fusion in their retrospective cohort studies.

Furthermore, in a randomized controlled trail (RCT), 80 patients were either enrolled into a standalone porous Ta cage fusion group or a pedicle screws-supplemented group [147]. Similar clinical evaluation results including the Oswestry Disability Index (ODI), VAS and SF-36 scores, were revealed at a 2-year follow-up point [147]. Furthermore, equivalent X-ray results of both groups evaluated at 6-year follow-up proposed that porous Ta standalone fashion could provide stability for lumbar spine interbody fusion without additional fixation or bone grafting [147]. However, RCTs are still rarely used to definitely corroborate the clinical value of porous Ta cages for lumbar spine fusion surgery used either in a standalone fashion or augmented with posterior screws. In addition, the radiopaque property of tantalum makes the radiographic examination of the intervertebral bone fusion rate difficult.

### 2.6. Ankle Arthrodesis and Arthroplasty

As with femoral head osteonecrosis, the end-stage ankle arthritis can also be a very severe and debilitating disease for younger and active patients [148,149]. Therefore, surgical intervention, e.g., ankle arthrodesis and total ankle arthroplasty, should be taken into consideration when conservative methods have failed.

Regarded as a promising alternative to traditional bone autograft or allograft, porous Ta spacer has been applied in ankle arthrodesis without the limits of size, volume and source [150,151,152,153]. Furthermore, the cost of a single porous Ta spacer (approximately $989.5–1000) has been reported to be approximately comparable to that of an iliac crest autograft (approximately $600–700) and an allograft (approximately $850); however, the latter two may take more time for preparation during the surgery [150,151,152]. The porous Ta spacer is an optimal choice for reconstruction surgery, and is especially suitable for huge bone defects [154,155]. This is the case because it has adequate structural strength to maintain the restored height and angular correction of the ankle joint until the appearance of osseous fusion between the porous Ta spacer and adjacent bony tissues [33,151], which is significantly different from bone autografts or allografts, either of which may collapse due to absorption after implantation [45,146,152]. Moreover, as with cancellous bone, the porous Ta spacer provides the necessary space and osteoconductive environment for vascularized bone tissue ingrowth, obviating autograft-related harvest lesions [156,157] and allograft-related infectious diseases [153].

The clinical results of porous Ta spacers used for the salvage of failed total ankle arthroplasty are also favorable [151,152,155]. More often, accompanied by nonunion, leg shortening, infection or even severe bone defect after debridement, failed total ankle arthroplasty can be difficult reasonably address [158,159]. To enhance the fusion efficiency of porous Ta spacer, Sundet et al. [160] combined the use of retrograde nailing, a porous Ta spacer and an osteoinductive pad augmented with autologous bone marrow concentrate for revision surgery of 30 patients (31 ankles) with failed total ankle arthroplasty. The mean fusion rate at the average 23-month follow-up was 93.5%, and the vast majority of patients were satisfied with the surgery in terms of pain relief and improved activity, though additional expenditure were entailed in this clinic trail [160]. Similarly, Kreulen et al. [155] introduced a new surgical strategy for reconstruction surgery of two patients with failed total ankle arthroplasty and four patients with ankle collapse post infection. In this study, porous Ta spacers were also augmented with autologous bone marrow obtained through the Reamer/Irrigator/Aspirator technique from the femoral marrow cavity and fixed with tibiotalocalcaneal nail, and the bone morphogenetic protein 2 (BMP-2) or platelet derived growth factor was further supplemented to boost bony fusion. With the help of this novel method, thorough osseous fusion at the implant–bone interface appeared at the early stage of 4–6 weeks post-surgery and no failure cases were observed [155]. In contrast, Aubret et al. [159] reported disappointing outcomes after the insertion of porous Ta spacers. Even augmented with iliac crest autograft and allograft bone chips for revision of failed total ankle arthroplasty in 10 patients, two patients had failed integration of porous Ta spacers, one patient presented with talocrural joint nonunion and three patients needed secondary revision surgery due to severe pain. However, the main reason for these failed cases was supposed to be the weak fixation strength provided by nails compared with 6.5 mm screws [151] or reconstruction plates [158].

Despite being reported as having a lower survival rate than hip and knee arthroplasty [161,162,163], total ankle arthroplasty (TAA) has been suggested to preserve the mobility of ankle joint and normal gait instead of being fused with triple arthrodesis which has long been considered as the gold standard for the treatment of end-stage ankle arthritis.

A newly designed porous Ta-based total ankle prosthesis was approved by the Food and Drug Administration in 2012 and marketed by Zimmer Biomet Inc. [164,165]. Combined with the use of porous Ta-based ankle prosthesis in TAA, promising prognosis can be foreseeable in terms of pain relief and functional improvement in the short-term, even without supplementation with cement augmentation, due to the fact that the stability of tibial and talar components mainly depends on bony interlocking between the porous Ta base and the host bone [164,165,166,167,168,169,170]. Moreover, the pattern of porous Ta bases of the two components resembles that of the subchondral bone of tibia and talus and can distribute loading stress rationally and diminish the occurrence of peri-implant osteolysis, which often resulted in aseptic loosening of the implants [165,171]. This novel ankle prosthesis is implanted through the lateral approach, associated with distal fibular osteotomy, which theoretically offers direct exposure to both the sagittal and coronal plane of the tibiotalar joint and obviates surgery-related neurovascular injuries [171]. Incorporated with an extramedullary alignment frame, the innovate surgery approach can minimize the amount of bony resection, optimize tibial and talar components positioning and preserve the bone–implant contact area, all of which finally guarantee the survival rate of porous Ta ankle prosthesis [164].

The histological analysis of this porous Ta-based ankle prosthesis retrieved from a 50-year old female patient revealed that the bone ingrowth percentage in tibial and talar components was more than those found in the retrieved porous Ta hip and knee components [172]. Meanwhile, active bone remolding was found within the porous Ta layer even at 3 years post-surgery. However, regional osteolysis and metal wear debris could not be avoided, both of which did not jeopardize the stability of the prosthesis. Nevertheless, decreased bone density of distal tibia adjacent to the tibial component still presented in this patient, indicating that the stress shielding effect and related bone resorption could not thoroughly be eradicated through the use of porous Ta-based ankle prosthesis [172].

### 2.7. Dental Implants

Aimed to increase surface energy, extend the bone–implant contact area, improve surface hydrophilicity and facilitate mesenchymal cells’ or osteoblast progenitor cells’ adherence, the surface roughness design of dental implants has now become very widely used and has been proven to enhance the progress of osteointegration and angiogenesis [173,174]. Therefore, the spongy bone like structure of porous Ta could be one explanation for its superior biological and mechanical property to many other metal materials in terms of rapid osseous ingrowth and bone-to-implant contact, both of which directly influence the survival rate of dental implants in the long run [175]. The histological and histomorphometric analysis has validated the osseoincorporation property of porous Ta implants derived from the rapid formation of vascularized bone tissues not only on the surface but also in the inner pores, which further reinforced the interlocking force between the implants and human jaws [176]. The canine model test revealed that the porous Ta section could provide a more rapid new bone formation and stronger stability for the porous Ta enhanced titanium implants compared to its conventional screwed titanium counterparts [177].

The porous Ta-enhanced tianium dental implant is now considered to be an effective therapeutic method for implanting treatment of certain patients associated with periodontitis [178], alveolar bone defects [179] and even maxillofacial tumors [180,181]. The porous Ta segment can provide an expanded three-dimensional space for the infiltration and differentiation of osteoblasts as well as the accumulation of vascular endothelial cells [40,182]. In addition, this novel implant has also been used in immediate revision surgery for previously failed dental implantation based on the superior osteointegration of porous Ta [183]. The immediate loading tests of porous Ta enhanced implants demonstrated significantly less marginal bone loss than that of threaded implants (0.43 ± 0.41 mm vs. 0.98 ± 0.67 mm) after 1-year of functional loading [184]. This result was then further corroborated in a retrospective study in which an average of 0.28 mm bone gain could be found in the porous Ta enhanced group, but the Ti group showed an average of 0.2 mm marginal bone loss after 1-year of implant loading [185].

However, mechanical flaw of this porous Ta enhanced dental implants may be located at the junction of the middle and distal third portion, for the middle portion is produced as slender sharp in order to accommodate the porous Ta sleeve and is welded to the distal apex portion [186]. Accordingly, potential fragile fracture may occur at this facet when the implant is to be inserted in the socket of maxilla or mandible with high bone density. Meanwhile, the unsterile oral cavity, where more than 500 kinds of bacteria are harbored, can be a challenge for the dental application of porous Ta [186]. Therefore, in-depth studies that can enhance the antibacterial property of porous Ta are still needed because the microbial environment of oral cavity and orthopedic sites is obviously different.

## 3. New Development of Porous Ta for Bone Tissue Engineering

### 3.1. Additive Manufactured Porous Ta

Except for conventional techniques including CVD [33,48], foam impregnation [49] and powder metallurgy [50], various additive manufacturing methods have been introduced to produce novel porous Ta scaffolds with different pore size and porosity, but comparable mechanical properties with human cortical and trabecular bones [47] (Table 1). Comparison tests performed with cellular and animal models have revealed similar or even better biological and mechanical performance of printed porous Ta scaffolds than their porous Ti counterparts with the same porosity and pore diameter (Table 2) [51,52,54,55,187]. Moreover, as a high-end technique, additive manufacturing can help manufacturers to produce porous Ta implants with tailored pore size and porosity to resist different biomechanical loading stress in different parts of the human body. Incorporated with Computer Aided Design (CAD) software, additive manufacturing thus makes personalized porous Ta implants or prostheses for individual patient possible. Recently, several printed porous Ta products have successfully been applied in clinical settings.

Wang et al. [188] have designed and produced a printed porous Ta knee prosthesis for revision surgery for an 83-year old female patient suffering from chronical inflammation and unendurable pain of the left knee after a previous total knee arthroplasty (Figure 5). The X-ray showed severe bone defect in the medial tibial plateau, varus deformity of the left knee and loosening of the tibial component, all of which were formidable challenges to be addressed by conventional surgical techniques. With the help of CAD, the authors corrected the anatomic alignment of the left lower limb and fabricated personalized knee prosthesis which can precisely match the bone defect area for the definite revision surgery. Twelve months after the final revision surgery, the patient recovered to normal activity with no more complaints about the affected limb. After that, the same team fabricated personalized porous Ta fibular and femur implants for reconstruction surgery following the same design and manufacturing process [53].

Developmental dysplasia of the hip (DDH) can lead to degenerative osteoarthritis of the hip in adults due to the malposition of acetabulum and femoral head [189]. In order to restore normal acetabular coverage of the femoral head and acetabulum index, the additive manufactured porous Ta acetabular patch was introduced in the treatment of eight adult DDH patients with Crowe type I [190]. Each individualized porous Ta acetabular patch was designed by Mimics 17.0 and 3-matic 9.0 software (Materialise, Leuven, Belgium) before surgery. Then, the loading stress distribution between the acetabulum restored by porous Ta patch and the femoral head was analyzed by Ansys 17.0 software (Ansys, Canonsburg, PA, USA). If the stress distribution was uniform, the designed porous Ta acetabular patch would be printed for the final surgery. After an average follow-up of 8.2 months, the VAS scores of eight patients were drastically decreased (2.92 ± 0.79 before surgery vs. 0.83 ± 0.72 after surgery). Meanwhile, the Harris scores (69.67 ± 4.62 before surgery vs. 84.25 ± 4.14 after surgery) and the results of gait analysis were greatly improved after the implantation of the porous Ta patch.

A printed porous Ta osteosynthesis plate has been used for the treatment of a 30-year old male patient with tibial nonunion [191]. The patient had undergone intramedullary nail fixation three times previously, but failed to attain healing even associated with the iliac crest autograft. Owing to its biological and biomechanical advantages, this novel porous Ta plate (80% porosity, 1.5–10 GPa elastic modulus) reunited the tibial shaft fracture uneventfully 5 months after the fourth surgery, and the patient regained normal mobility (Figure 6).

Nevertheless, the high demand and high price of the medically applicable tantalum powder used to produce porous Ta products are the main negative factors that hinder the extensive clinical implementation of novel porous Ta implants or prostheses.

### 3.2. Surface Modification

The critical drawbacks that may impede the further application, in bone tissue engineering, of porous Ta are its inertness and low level of bioactivity. Therefore, various methods have been introduced to modify porous Ta for further clinical application (Table 3). These methods can mainly be cataloged into biomaterial coating and surface treatment, all of which are aimed to endow porous Ta-based implants or prosthesis with improved osteoconductivity, osteoinductivity and antibacterial properties (Figure 7).

Calcium phosphate (CaP) and hydroxyapatite (HA) are not only the mineral components of human bones, but have also been exploited in porous Ta modification for surface modification and drug delivery [200,201,202]. Furthermore, the alendronate–CaP coated porous Ta has been verified to fill the bone–implant interface gaps, with an average length of 0.6 mm, in rabbit models after 4 weeks [200]. The mechanism behind this successful restoration of simulated bone defects could be attributed to the slowly released alendronate, which inhibited the activity of osteoclasts but enhanced that of osteoblasts at the same time. Similarly, the zoledronic acid-HA coated porous Ta rod also gained significantly more bone formation both at the peri-implant area and within the inner space compared with the unmodified porous Ta groups in canine models [201]. Zhou et al. introduced amorphous calcium phosphate (ACP) nanosphere and HA nanorod coating to modify porous Ta [192]. When immersed in SBF, the two nanostructures showed rapid mineralization on their surface and the mineral deposition increasingly accumulated within 1 week. Simultaneously, the hydrophilicity of two structures was also significantly improved due to the capillary effects. The ACP nanospheres were observed to transform into HA nanosheets in a rapid pace after being soaked in SBF, and this transformation promised rapid mineralization, improved wettability and faster protein release rates [192,193]. In vivo, both kinds of modified porous Ta scaffolds repaired the subchondral bone defects with substantial new bone formation, indicating a promising clinical prospect for bone defect restoration.

Bone morphogenetic protein 7 (BMP-7) has been applied in bone and cartilage repair since 2001 due to its powerful osteoinductivity [203,204,205]. BMP-7 can act as a bone stimulating agent that induces differentiation of mesenchymal stem cells into osteoblasts and chondroblasts [206]. By soaking porous Ta in the solution of BMP-7, Wang et al. [194] coated BMP-7 on the surface of porous Ta rods. Subsequently, the BMP-7 modified porous Ta rods obtained satisfactory results of subchondral bone and cartilage repairing in rabbit models with substantial chondroid-like tissues recovering in the defect areas within 16 weeks. Furthermore, bone ingrowth depth was found to be 0.2–1.2 mm in the modified samples, which finally resulted in rigid bone–implant interlocking.

Fabrication of Ta_2_O_5_ nanotube layers on the surface by anodization [195,196] or micro-arc oxidation (MAO) [207] is another approach to ameliorate the bioactivity of Ta. With the formation of nanotubes, the Ca and P elements contained in electrolytes can be incorporated into the oxide nanotubes by either of the aforementioned methods [208]. However, MAO may result in toxic effects on cell viability due to the by-products, i.e., reactive oxygen species (ROS) and reactive nitrogen species (RNS). Combined with alkali pretreatment, these toxic elements produced by the process of MAO were dissolved and the newly formed sodium tantalate layer and could further facilitate the deposition of apatite in SBF. It is well defined that the substantial apatite layer formed on the surface of implants is the prerequisite for bone–implant integration [19,197]. In this regard, the combination of MAO and alkali treatment will be an effective way to modify porous Ta to boost its osteoconductivity.

Implant-associated infection has long been a thorny problem in clinical settings, which always results in catastrophic failure and additional expenditure [209,210]. It is imperative to find rational methods to endow porous Ta with antibacterial property. Polyhydroxyalkanoates (PHAs) are biodegradable and biocompatible materials which can be used as natural carrier for drug delivery and scaffold for tissue replacement [211]. Loading PHAs coating containing antibiotics on the surface of porous Ta and obtaining a controlled antibiotics release will be an optimal choice to avoid implant-associated infection [212]. Rodríguez-Contreras et al. [198] coated the PHA–Genta composite layer both on the outer and inner surface of porous Ta cervical fusion cages. The continuously released Genta from PHA coating with homogeneous concentration protected these porous Ta cages from infection of Gram^+^ and Gram^−^ bacteria. On the other hand, a ZnO nanorod–nanoslice hierarchical coating was proposed by Liao et al. [199]. In vitro, the ZnO nanoslice was first released from the superficial layer to kill bacteria during the early stage, and the antibacterial efficacy lasted for 24 h. By contrast, the release of ZnO nanorod showed a slow but stable pattern. Therefore, the combined ZnO nanorod–nanoslice coating possessed a two-stage release pattern and could last for over 2 weeks in vivo, avoiding the implant-associated infection which commonly occurred within 1 week post-surgery [199].

## 4. Conclusions

Owing to its excellent biological and mechanical properties, porous Ta is an optimal biomaterial for bone tissue engineering and has gained satisfactory clinical results, though modifications are needed to refine it. With the advent of additive manufacturing, the printed porous Ta has shed light on the design and manufacture of novel porous Ta-based implants for individualized healthcare as the macrostructure, pore size, pore geometry and porosity of porous Ta implants can be adjusted to meet the needs of the host, especially when huge and complex bone defects are present at the load-bearing parts. Moreover, various modification methods have been emerged to enhance the bioactivity and antibacterial activity of porous Ta. In addition, the modified porous Ta will definitely be used to cope with various pathological conditions, e.g., osteoporosis, infection, diabetes and even tumors. However, in-depth studies are still desired to explore the potential development of porous Ta. Firstly, the impact of the topological structure on the biological and mechanical properties of porous Ti or Ti alloy has been fully detected, but the relative research on porous Ta is rare. Since Ta has entirely different characteristics compared with Ti or Ti alloy, porous Ta scaffolds with different topological macro- or micro-structures should be determined to verify their biological and mechanical properties and for further application in different biological and mechanical environments. Secondly, abundant randomized controlled clinical trials (RCT) with sufficient samples and long-term follow-up are still desired to further verify the clinical practicability of modified porous Ta implants. At present, the clinical application of additive manufactured porous Ta implants is mainly confined by the high price of printing individual porous Ta implants. With the development of additive manufacturing technology and the expansion of the additive manufacturing market size, the price of printed porous Ta will decrease sooner or later, and the extensive application of printed porous Ta implants is thus on the horizon, given that the prevalence of aging populations entails increases in orthopaedic arthroplasty and dental implantology.

## Figures and Tables

**Figure 1 materials-14-02647-f001:**
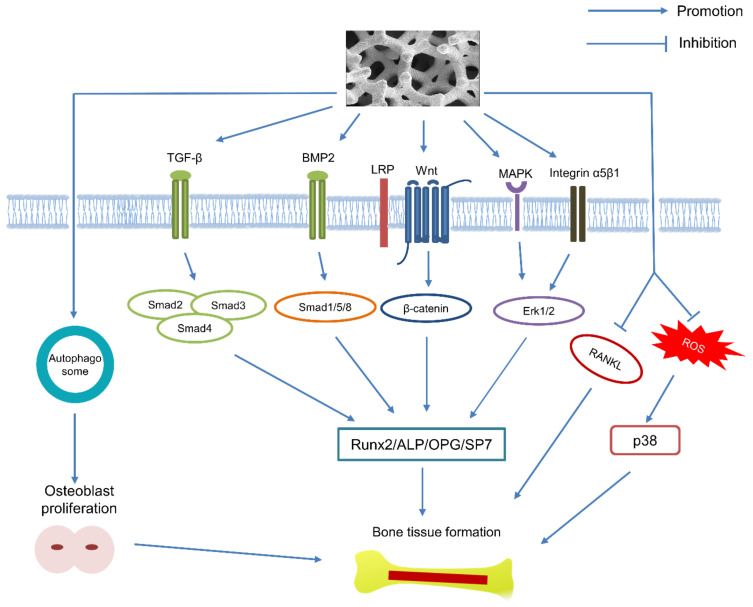
Schematic representation of the relative signaling pathway that may be involved in the osteogenic effect of Ta.

**Figure 2 materials-14-02647-f002:**
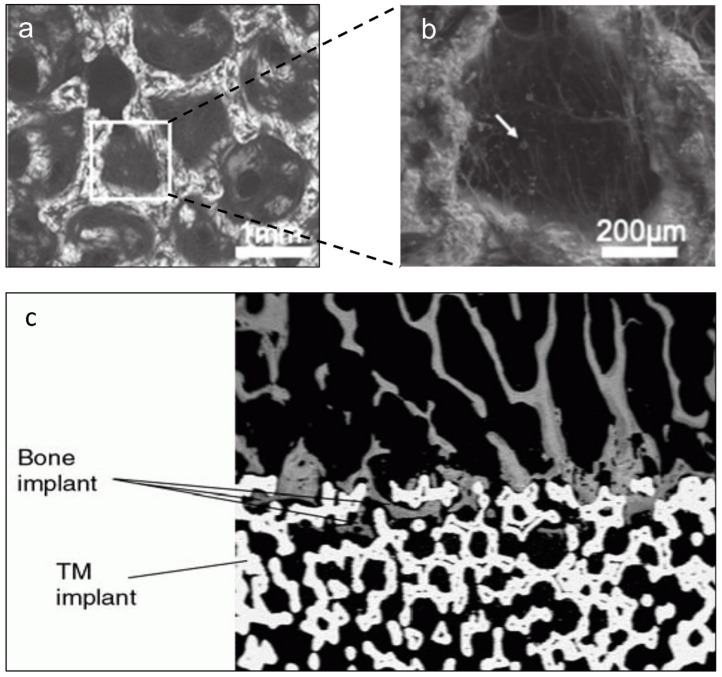
The microstructure of porous Ta presented as honeycomb structure (**a**), and cells that partially cover the cavity with many calcium nodules (indicated with white arrow) can be detected (**b**). Reprinted from ref. [36]. Abundant bone ingrowth can be found in the pores of porous Ta implant (**c**). Reprinted with permission from [45]. Copyright © 2021 by American Academy of Orthopaedic Surgeons.

**Figure 3 materials-14-02647-f003:**
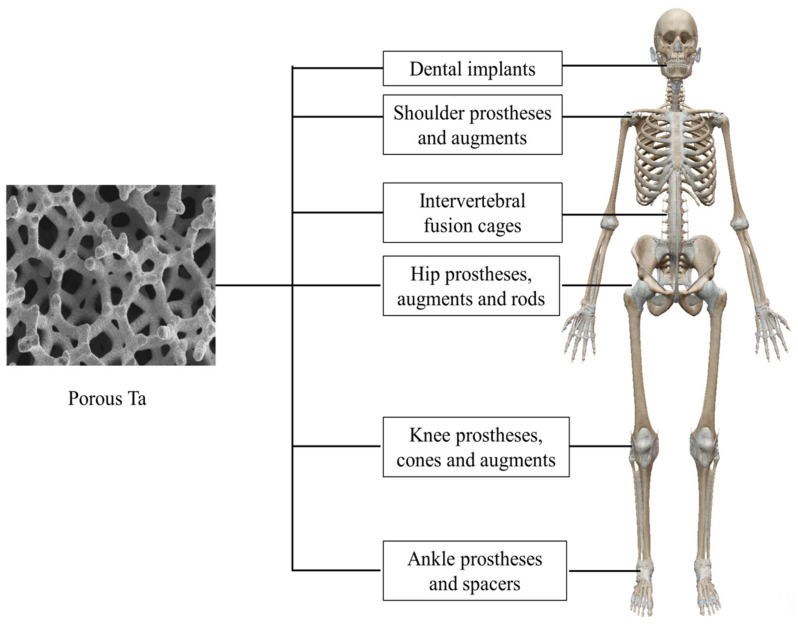
Application of porous Ta in different parts of the human body.

**Figure 4 materials-14-02647-f004:**
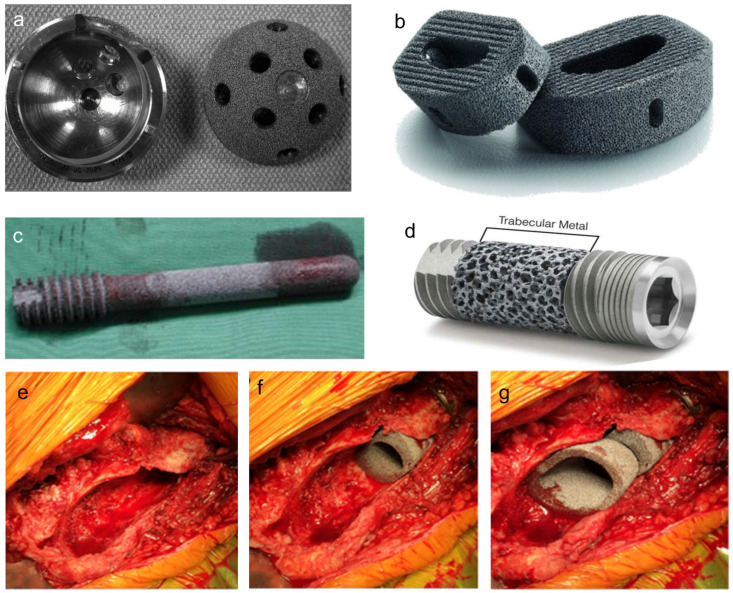
The typical products of porous Ta-based implants manufactured by Zimmer Biomet Inc. Acetabular cup with porous Ta coating (**a**). Reprinted with permission from [45]. Copyright © 2021 by American Academy of Orthopaedic Surgeons. Porous Ta lumbar interbody fusion cage (**b**) Reprinted from ref. [57], porous Ta rod (**c**) Reprinted from ref. [58] and dental implant (**d**) Reprinted from ref. [59]. The porous Ta cones were used to reconstruct femoral metaphyseal defect (**e**–**g**). Reprinted from ref. [60].

**Figure 5 materials-14-02647-f005:**
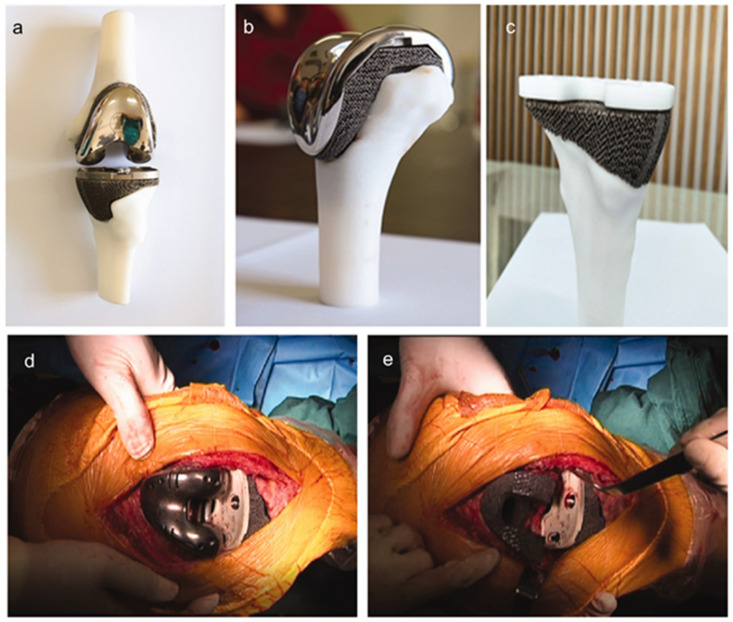
The printed personalized porous Ta knee prosthesis (**a**), distal femoral component (**b**) and proximal tibial component (**c**). The porous Ta prosthesis was inserted into distal femur and proximal tibia, respectively, during the surgery (**d**,**e**). Reprinted from ref. [188].

**Figure 6 materials-14-02647-f006:**
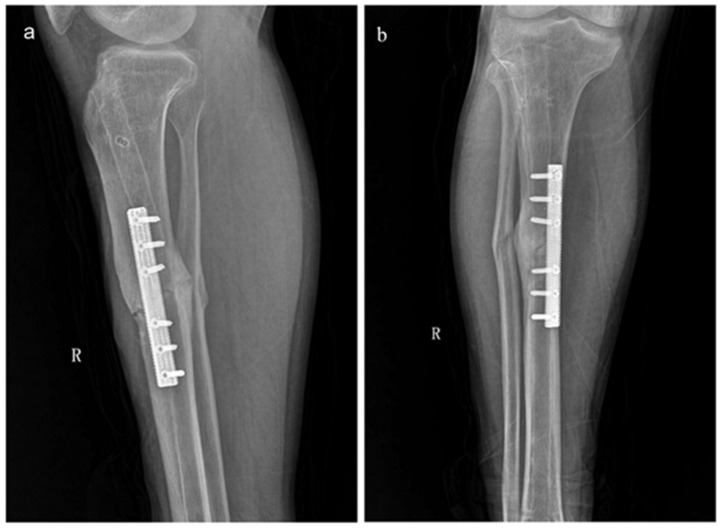
The AP (**a**) and lateral view (**b**) of X-ray examination at 5-month follow-up showed that the fracture healed after the implantation of the printed porous Ta osteosynthesis plate. Reprinted from ref. [191].

**Figure 7 materials-14-02647-f007:**
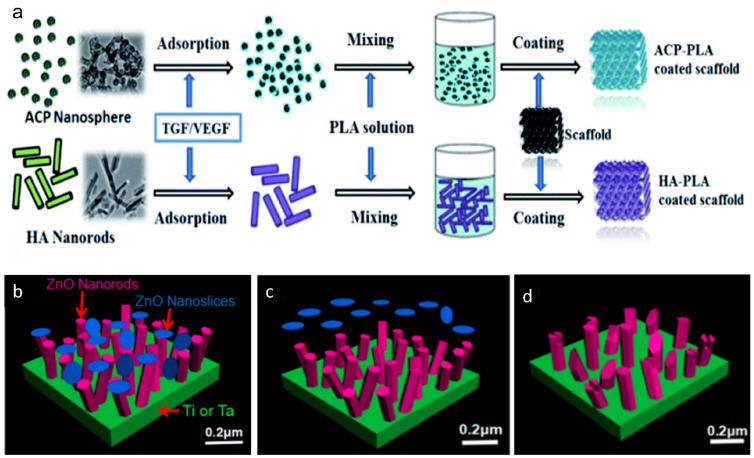
Schematic diagram of the surface modification for porous Ta. Amorphous calcium phosphate (ACP) nanospheres and HA nanorods coating on the surface of Ta scaffold (**a**). Reprinted from ref. [192]. ZnO nanoslices and ZnO nanorods coating on Ta substrate (**b**), the ZnO nanoslices will be released at an early stage—within 48 h (**c**), while the ZnO nanorods are released in a slow pattern over 2 weeks (**d**). Reprinted with permission from [199]. Copyright © 2021 by American Chemical Society.

**Table 1 materials-14-02647-t001:** The mechanical properties of osseous tissues and porous Ta produced by different techniques.

Osseous Tissues	Manufacturing Technique	Porosity (%)	Pore Size (μm)	Strut Size (μm)	Elastic Modulus (GPa)	Compressive Strength (MPa)	Yield Strength (MPa)	0.2% Proof Strength (MPa)	Ref
Cortical bone		3–5			7–30	100–230			[47]
Trabecular bone		50–90			0.01–3.0	2–12			
	CVD (porous carbon scaffold)	75–85	400–600	40–60	2.5–3.9	42–78			[33]
	CVD (porous SiC scaffold)	70–85	150–400	40–60	10–30	35–100			[48]
	Foam impregnation	65–80	400–600		2.0–4.6	100–170			[49]
	Powder metallurgy		100–400		2.0 ± 0.3	50.3 ± 0.5			[50]
	LENS	55			1.5 ± 0.3			100 ± 10	[51]
		45			7 ± 0.6			192 ± 7	
		27			20 ± 1.9			746 ± 27	
	SLM	80	500	150	1.22 ± 0.07	28.3 ± 1.2	12.7 ± 0.6		[52]
	SEBM	75		540			23.98 ± 1.72		[53]
		80		392			19.48 ± 1.45		
		85		386			6.78 ± 0.85		
	SLM	70	500	400	3.10 ± 0.03				[54]
	SLM	80	300–400		2.34 ± 0.2	78.54 ± 9.1			[55]

Notes: CVD, Chemical Vapor Deposition; LENS, Laser Engineered Net Shaping; SLM, Selective Laser Melting; SEBM, Selective Electron Beam Melting.

**Table 2 materials-14-02647-t002:** The biological properties of additive manufactured porous Ta scaffolds.

Porosity%/Samples	In Vitro Tests Results	In Vivo Tests Results	Ref.
80% Ta	**Cytotoxicity test** (L929 mammalian cells) No cytotoxicity	**Histological evaluation** (rat femur defect model) The bone defect can be bridged by the new bone with the help of printed porous Ta scaffold. **Torsion test** Rigid bone–implant connection can be obtained.	[52]
70% Ta vs. 70% Ti	**Cell morphologies** (hBMSCs) Cells’ adhesion, proliferation and vitality were similar. **Cell differentiation** ALP and mineralized nodule staining levels were comparable. **Quantitative RT-PCR Analysis** Sp7 and OCN genes levels were comparable.	**Histological evaluations** (rabbit distal femoral defect model) Bone ingrowth rate and depth were similar in the two groups. Ti group showed a quick-slow-quick new bone formation pattern. Ta group showed a gradual slowdown style of new bone formation. **Push out test** The two groups had similar push out force.	[54]
80% Ta vs. 80% Ti	**Cell morphologies** (hBMSCs) Ta group showed better cell viability than Ti group. **Cell proliferation** Ta group was higher than Ti group after 5–7 days. **Cell differentiation** Ta group had superior ALP levels and calcium nodule numbers. **Quantitative RT-PCR Analysis** Levels of Runx2, ALP, Col-1, OCN and OPN genes were higher in Ta group.	**Histological evaluations****and f****luorescence labeling** (rabbit distal femoral defect model) Ta could stimulate new bone formation as early as 4 weeks.	[55]
30% Ta vs. 30% Ti modified with TiO_2_ nanotubes, 30% Ti and solid Ti	Not mentioned	**Histological analysis** (rats distal femur model) Ta group had the most significant bone formation after 12 weeks. **Push out test** Four groups had similar bone–implant interlocking strength. **FESEM micrographs** Ta groups had persistent bone ingrown in the pores at 12 weeks. Ti modified with TiO_2_ nanotubes groups showed comparable seamless bone–implant interface with Ta groups. The other two Ti groups had inferior bone–implant contact.	[187]
27% Ta and 45% Ta vs. 27% Ti	**Cell morphologies** (hFOB CRL-11372) Ta groups presented more flattened cell morphologies, filopodia extensions and mineralization than Ti group. **Cell proliferation** Cells proliferated rapidly on Ta samples instead of Ti samples. **Immunochemistry** Porous Ta facilitated cells’ adhesion and differentiation via a porosity-dependent pattern.	Not mentioned	[51]

Note: FESEM, field emission scanning electron microscope; hBMSCs, human bone mesenchymal stem cells.

**Table 3 materials-14-02647-t003:** The biological performance of different methods for Ta modification.

Surface Modification	In Vitro Test Results	In Vivo Test Results	Ref.
ACP nanospheres–PLA coating HA nanorods–PLA coating	**Mineralization in SBF** Abundant mineral deposition could be formed in 1 week. **Hydrophilicity** After being soaked in SBF for 1 day, the hydrophilicity of the two coatings was improved. **Protein adsorption and release** The two nanostructures possessed satisfactory VEGF-FITC adsorption. The amount of BSA release from ACP nanospheres–PLA coating was faster and larger. **Cell viability and morphology** (MG63 cells) The two nano-coatings showed no toxic effects on cells. Cells’ adhesion, interconnecting and spreading were better than those cultured on unmodified samples.	**Subchondral bone defect repair** Significant new bone formation could be found in samples modified by two coatings. By contrast, new bone tissues were lacking in the unmodified samples.	[192]
CaP nanospheres–PLA coating	**Mineralization in SBF** CaP nanospheres coating transformed into HA nanosheet which could continuously accumulate on the surface of Ta. **Hydrophilicity** CaP nanosheres–PLA coating showed satisfactory hydrophilicity. **BSA release** The transformation from amorphous CaP to HA induced the rapid release of BSA at an early stage. **Cell viability** (MG63 cells) Cells established fine adhesion to CaP nanosheres–PLA coating.	**Subchondral bone defect repair** The modified porous Ta scaffold effectively repaired the defect after 12 weeks.	[193]
BMP-7 coating	Not mentioned	**Cartilage defect restoration** (rabbit model) Modified porous Ta significantly facilitated cartilage restoration at 4, 8 and 16 weeks. **Microscopic and histological analyses** Modified porous Ta groups facilitated calcium salt deposition, as well as formation and maturity of bone and cartilage tissues. **Micro-CT analyses** Sixteen weeks post-surgery, new bone formation could be found around the modified porous Ta. The amount of new bone formation was more than those of unmodified samples. **Push out tests** The modified groups possessed higher maximum push out force.	[194]
Ta_2_O_5_ nanotubes films	**Anticorrosion test** Ta_2_O_5_ nanotube films had excellent biocompatibility and prevented the release of ions. **Contact angle and surface energy** Wettability and surface energy of Ta were enhanced by Ta_2_O_5_ nanotube films. **Protein adsorption** Adsorption of BSA and Fn were significantly more on Ta_2_O_5_ nanotube films than bare surface, **Cell adhesion and proliferation** (rBMSCs) Adhesion and proliferation of rBMSCs were highly enhanced on Ta_2_O_5_ nanotube films. **Osteogenesis-related genes expression** Levels of Osterix, ALP, Collagen-I and Osteocalcin were significantly high on the Ta_2_O_5_ nanotubes films. **Fluorescence microscopy image** Cells cultured on Ta_2_O_5_ films presented as polygonal morphology and had more filopodia than those on bare surface.	Not mentioned	[195]
Nanoporous Ta oxide layers	**Cell proliferation and morphology** (L929 mouse fibroblasts) Nanoporous Ta oxide layers with 25 nm pore size greatly enhanced adhesion, proliferation and extension of fibroblasts.	Not mentioned	[196]
MAO combined with NaOH treatment	**Mineralization in SBF** Substantial mineral deposition can be found on the surface of porous Ta treated with MAO and NaOH etching. **Cell proliferation** (3T3-E1 cells) Cell proliferation on the modified samples was better than the untreated ones at 24, 48 and 72 h. **Cell morphology** Cells spread over the surface and migrated into the pores of the modified samples, with increasingly filiform protrusions and calcium crystals presented.	**Bone ingrowth** (rabbit cranial defect model) New bone formation could be found around the modified samples at 4 weeks. Bone remolding and neovascularization were also found within the pores. The cranial defect could be filled by new bone at 12 weeks.	[197]
PHAs (PHB, PHBV and PHB4HB)–Genta coating	**Cytotoxicity and cell adhesion** (SaOS-2 cells) PHAs coating showed no toxicity to the cells. **Antibacterial properties** (*S. aureus* and *E. coli*) The concentration of Genta released from PHAs coating effectively inhibited the proliferation of *S. aureus* and *E. coli*.	Not mentioned	[198]
ZnO nanorods−nanoslices hierarchical structure coating	**Antibacterial Properties** (*S. aureus* and *E. coli*) The novel ZnO coating showed a two-stage release pattern and effective antibacterial properties. **Cytotoxicity** (MC3T3-E1 cells) The ZnO nanorods–nanoslices coating had no toxic effect on cells.	**In vivo Infected Studies** (KM mice subcutaneous implantation) The ZnO nanorods–nanoslices coating modified Ta foils had ideal antibacterial performance which could last for over 2 weeks in vivo.	[199]

ACP: amorphous calcium phosphate; HA: hydroxyapatite; PLA: polylactic acid; SBF: simulated body fluid; PHAs: polyhydroxyalkanoates; Genta: gentamicin sulfate; BMP-7: bone morphogenetic protein 7; BSA: bovine serum albumin; Fn: fibronectin; rBMSCs: rabbit bone mesenchymal stem cells; BSA: bovine serum albumin; CaP: calcium phosphate; MAO: micro-arc oxidation; *E. coli*: Escherichia coli; *S. aureus*: Staphylococcus aureus.

## Data Availability

No new data were created or analyzed in this study.

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
