# Peer review of "The Clinical Application of Porous Tantalum and Its New Development for Bone Tissue Engineering"

_materials, 2021, doi:10.3390/ma14102647_

Round 1

Reviewer 1 Report

The manuscript materials-1196895 The clinical application of porous Tantalum and its new development for bone tissue engineering; is a review article on the application and behaviour of porous tantalum materials in biomedicine (implantology, dentistry, bone tissue engineering etc.). Porous Ta is one of the youngest material used in the biomedicine sector where corrosion resistance and good strength has to be supported. It has high wettability and surface energy which facilitates the adherence and proliferation of osteoblasts. Its low elastic modulus and high friction coefficient avoids the stress shield effect, and ensure the primary stability of the bones. Authors have confirmed that there are many porous and solid Ta applications in biomedicine and compared the results with other materials, i.e. porous and solid titanium. The manuscript also showed the review of very interesting modification methods of the surface for better bioactivity and cytotoxicity of Ta, which is not always satisfactory, and here I think it is a very crucial section of the manuscript.  Other areas of the article are also extraordinarily vital. Section 2 Authors have presented the tantalum results in different clinical applications in implantology as hip arthroplasty, knee arthroplasty, shoulder arthroplasty, ankle arthrodesis, and arthroplasty, spine implants, dental implants. I appreciate showing these research results, but I think it would also be worth offering some images of such implants or applications. Section 3 of the article describes and reviews the new ways in the manufacturing and developing of porous Ta for bone tissue engineering by using different additive manufacturing methods (SLM, LENS) and also conventional methods (PM, CVD). In this section, the Authors also put together new methods for Ta surface modifications and showed the in-vitro and in-vivo results of the biological performance of the modified Ta. It has good scientific value to systematizing the methods and guide their advantages in tailoring the implants. Nevertheless, the Authors should also improve section 3 with some more images showing the surface results, maybe macro- or microstructural observations of the Ta surface. Such presentation would give a much deeper insight into the porous Ta layers properties, structure and type. Summing up, I can state the Authors reached the aim of the investigation. The presented research is vital for biomedical applications of porous Ta in many sectors (orthopaedics, dentistry etc.). The compilation of the results obtained for porous Tantalum in perspective will give the Authors and engineers possibilities to produce the novel implants and biomaterials. However, the insufficient amount of images with examples of Ta application is noticeable here. The manuscript has high scientific value but should be upgraded to attract the Readers' attention in such a form. As I read the manuscript, I also found some mandatory errors that the authors should correct before the publication. I recommend performing minor revisions of the manuscript. Some detailed suggestions are listed in the following points:

1. The conclusions in the article are too general.
I suggest re-editing the conclusions and extending the description of the impact of the porous Ta materials on novel implants and biomaterials. I think The Authors should also highlight the main outlook and applications for the future here.

2. Some sentences are unclear, please re-edit the fragments and sentences:

Line 88: is – ,and selective lase melting (SLM).

    should be – ,and selective laser melting (SLM).

Line 230: is – approximately $4,000)

      should be – approximately $4.000)

Line 434: is – now gone virus 

      should be – now gone viral

Line 482: is – Compute Aided Design

               should be – Computer Aided Design

Line 219: happen.. should be ...happen.

3. Please add space between the units:

Line 60, 600 µm

Line 308, 2-3 mm

Line 400, 6.5 mm

Line 526, 1.5 – 10.0 GPa

Line 548, 0.6 mm

4. Please add space between the sentence and reference:

Lines 51, 52, 64, 71, 84, 90, 94, 376, 509, 512, 546, 548, 555

5. Figure captions: please rewrite the caption of Figure 1.

Figure 1:

Porous Ta has been extensively applied in different parts of human body for 103 restoration surgery due to its superior biological and mechanical properties.

i.e. Figure 1. Application of porous Ta in different parts of the human body.

The Authors should improve the picture with more images of the implants, surface modifications etc., in such a form it has low quality of presentation and scientific value.

There are no more comments that I felt to comment on. In general, the article requires editorial refinement and proofreading. The issues presented in the article are novel, significant and suitable for publication in the Journal Materials, but to increase their scientific value article requires the addition of the images and should be again analyzed by the Authors. I also want to point out that the Authors should format the manuscript according to the journal's and the MDPI publishing house guidelines. I recommend the paper for publication in Journal Materials after minor revisions.

Author Response

Dear Reviewer:

Thank you for your comments concerning our manuscript.

We have studied comments carefully and have made correction which we hope meet with approval. Revised portion are marked in red in the paper. The main corrections in the paper and the responds to the reviewer’s comments are as flowing:

  1. The conclusions in the article are too general.

We have re-edit the conclusion portion and focus on the future application and research direction of porous Ta. We also highlight the individualized healthcare value of additive manufactured porous Ta as well as the modified Ta implants for various pathological condition.

  1. Some sentences are unclear, please re-edit the fragments and sentences.

We have re-edit the sentences in each paragraph.

  1. Please add space between the units.

Spaces have been added between each unit.

  1. Please add space between the sentence and reference.

Spaces have also been added.

  1. Figure captions: please rewrite the caption of Figure 1.

The figure legend has been revised.

  1. The Authors should improve the picture with more images of the implants, surface modifications etc.

There are 7 pictures in the revised manuscript. And we believe the new added picture will enrich our manuscript.

Reviewer 2 Report

The manuscript is a well-documented review consisting of two major parts: one related to seven clinical applications of porous tantalum and the second related to some recent developments (additive manufacturing and surface modification). The manuscript is easy to read, providing sufficient information for a facile understanding of the topic.

I have the following suggestions:

  • Please consider adding one or more figures (such as more schematic representations of the concepts presented in the text) to improve the visual impact of your manuscript.
  • Please discuss in more detail the specific research gaps related to the use of porous tantalum and some possible research directions/research trends that may address those gaps.

Author Response

Dear reviewer :

Thank you for your nice comments.

We have corrected the manuscript, according to recommendation, with addition/corrections in red in the revised manuscript.

Here are also some elements of answer :

  1. Adding one or more figures (such as more schematic representations of the concepts presented in the text) to improve the visual impact of your manuscript.

We have revised the figures and figure legend. Now, there are 7 figures in the revised manuscript, and we believe the new added figures can enrich our manuscript.

  1. Please discuss in more detail the specific research gaps related to the use of porous tantalum and some possible research directions/research trends that may address those gaps.

The relative gaps and research direction has been added in conclusion portion.

Reviewer 3 Report

The author presents a valuable review regarding the potential application of the porous Ta as bone implants. The clinical information is relevant and could provide a broad look at the possibilities to applied porous Ta to treat several diseases. 
However, as the clinical overview exposed by the authors have significant value, it is necessary to highlight the aspect, properties, and functionalities of the Ta as material. No significant data or information regarding the Ta performance as biomaterials has been provided.
The manuscript must include Ta microstructures, and graphs, to show the microstructure-performance relationships and the mechanical behavior as a function of the pore size, porosity percentage, and culture environment, etc.
The authors have mentioned only one company that manufactured Ta implants? Where are those others implant made from? I full list of companies must be included.
Many of the clinical applications mentioned by the authors are undergoing, therefore the material performances under such conditions need to be addressed. 
Sadly, I must recommend to not publish this review in the present form, the material aspect of the porous Ta must the included and discussed in the manuscript.

Author Response

Dear Reviewer:

Thank you for your comments concerning our manuscript entitled.

Those comments are all valuable and very helpful for revising and improving our paper. Revised portion are marked in red in the paper. The main corrections in the paper and the responds to the reviewer’s comments are as flowing:

  1. The manuscript must include Ta microstructures, and graphs, to show the microstructure-performance relationships and the mechanical behavior as a function of the pore size, porosity percentage, and culture environment, etc.

The relationship between microstructure and mechanical performance as weel as biological behavior has been discussed in paragraph 3 and 4. And relative data has been illustrated in Table 1.In addition, figure 2 has shown the microstructure of porous Ta, osteogenic differentiation of stem cells in the cavity and bone ingrowth in the pores of porous Ta.

  1. The authors have mentioned only one company that manufactured Ta implants? Where are those others implant made from?

The other two companies have been added in paragraph 5.

Round 2

Reviewer 1 Report

The Authors of the review manuscript no. 1196895 The clinical use of porous tantalum and its new development in bone tissue engineering, have realized most of the corrections suggested by the reviewer. The Authors have amended the Introduction section of the manuscript, and as a result, it sounds better and more accurate when it comes to the application of porous tantalum. The Authors added Figures 1, 2 to better illustrate Ta materials' osteogenic effect, and Figure 4 showing some Ta-based implants.  Section 3. The new design of porous Ta for bone tissue engineering has also been greatly improved and corrected according to my suggestions. Figure 7, which describes the modification of the porous Ta surface is significant here.  The Conclusions section has also been improved by summarizing the research and highlighting perspectives. The Authors of the article fully described the research on tantalum as a biomaterial and finally proved that it is one of the promising materials for implantology. I think the Authors have corrected and proofread the article according to my recommendations, which I really appreciate. Sections and figures added by the Authors in the manuscript are essential and interesting. The issues presented in the article are novel and suitable for publication in the Journal Materials. Nevertheless, I still urge the Authors to format the manuscript text and references according to the journal's editorial guidelines. I recommend the article for publication in Journal Materials in present form. 

Reviewer 2 Report

I have read the revised manuscript and the authors' response. I consider that my queries have been addressed and I have no additional suggestions

Reviewer 3 Report

The revised version provided by the authors addresses the main concert and suggestion rised. 

The incorporation of figures showing Ta microstructure, and adding more details about the material, processing, and performances, have improved the quality of this review.